# Coating with Hypertonic Saline Improves Virus Protection of Filtering Facepiece Manyfold—Benefit of Salt Impregnation in Times of Pandemic

**DOI:** 10.3390/ijerph18147406

**Published:** 2021-07-11

**Authors:** Franz Tatzber, Willibald Wonisch, Gyula Balka, Andras Marosi, Miklós Rusvai, Ulrike Resch, Meinrad Lindschinger, Sabrina Moerkl, Gerhard Cvirn

**Affiliations:** 1Otto Loewi Research Center, Division of Immunology and Pathophysiology, Medical University of Graz, 8010 Graz, Austria; franz.tatzber@medunigraz.at (F.T.); sabrina.moerkl@medunigraz.at (S.M.); 2Otto Loewi Research Center, Division of Physiological Chemistry, Medical University of Graz, 8010 Graz, Austria; gerhard.cvirn@medunigraz.at; 3Department of Pathology, University of Veterinary Medicine, 1078 Budapest, Hungary; Balka.Gyula@univet.hu; 4Department of Microbiology and Infectious Diseases, University of Veterinary Medicine, 1143 Budapest, Hungary; marosi.andras@univet.hu (A.M.); Rusvai.Miklos@univet.hu (M.R.); 5Department of Vascular Biology and Thrombosis Research, Medical University of Vienna, 1090 Vienna, Austria; uresch@gmx.at; 6Institute of Nutritional and Metabolic Diseases, Outpatient Clinic Laßnitzhöhe, 8301 Laßnitzhöhe, Austria; meinrad@lindschinger.at; 7Department of Psychiatry and Psychotherapeutic Medicine, Medical University of Graz, 8036 Graz, Austria

**Keywords:** SARS-CoV-2, hypertonic saline, COVID-19, protection, transmissible gastroenteritis virus, alphacoronavirus 1

## Abstract

Recently, as is evident with the COVID-19 pandemic, virus-containing aerosols can rapidly spread worldwide. As a consequence, filtering facepieces (FFP) are essential tools to protect against airborne viral particles. Incorrect donning and doffing of masks and a lack of hand-hygiene cause contagion by the wearers’ own hands. This study aimed to prove that hypertonic saline effectively reduces the infectious viral load on treated masks. Therefore, a hypertonic salt solution´s protective effect on surgical masks was investigated, specifically analyzing the infectivity of aerosolized Alphacoronavirus 1 in pigs (Transmissible Gastroenteritis Virus (TGEV)). Uncoated and hypertonic salt pre-coated FFPs were sprayed with TGEV. After drying, a defined part of the mask was rinsed with the medium, and the eluent was used for the infection of a porcine testicular cell line. Additionally, airborne microorganisms´ long-term infectivity of sodium-chloride in phosphate-buffered saline comprising 5% saccharose was investigated. In the results from an initial Median Tissue Culture Infectious Dose, infection rate of TGEV was minimally reduced by untreated FFP. In contrast, this could be reduced by a factor of 10^4^ if FFPs were treated with hypertonic salt solutions. Airborne pathogens did not contaminate the growth medium if salt concentrations exceeded 5%. We conclude that hypertonic saline is a vital tool for anti-virus protection, exponentially improving the impact of FFPs.

## 1. Introduction

The severe acute respiratory syndrome coronavirus 2 (SARS-CoV-2) provoked the coronavirus disease 19 (COVID-19) pandemic and is still spreading globally [1]. Even after effective vaccines or antivirals are made available, alternative approaches are required to prevent infection and decrease the alarmingly high mortality rate. One essential prevention measure is the mandatory wearing of a filtering facepiece (FFP), which proved to be an efficient prevention strategy against SARS-CoV-2 [2]. Wearing masks is inevitable for infected patients and healthy persons who are in close contact scenarios. However, several points must be followed: masks need correct adjustment, the outer layer should not be touched, and surgical masks should be used only once. An extended use provokes discomfort and a diminished effect due to the accumulation of pathogens; hence, the risk of infection may be critically increased. Furthermore, FFPs may provide a false sense of security as they do not protect the wearer. In addition, compliance with appropriate utilization is low, as suggested by the World Health Organization. This is evident even in Japan, where the wearing of face masks is a kind of cultural normality. Machida et al. [3] reported that only 23.1% of people complied with all recommendations. Another study indicated that only 6.8% of healthcare workers in Poland were compliant with the need to avoid touching the mask with their hands [4]. Previously, an extraordinarily effective and simple solution to most of these problems was published [5,6], i.e., the coating of FFPs with hypertonic saline. After mask impregnation with hypertonic sodium chloride (NaCl) solutions, the salt crystallizes while drying on the cloth surface. These salt crystals are hygroscopic and attract aerosol water droplets. This leads to the inactivation of microorganisms present in such droplets due to aerosol contact with NaCl. In the case of virus particles, surface proteins become denatured, and thereby lose their structural integrity. This prohibits binding to receptor cells, introducing the infection. In the 1960s [7], a study revealed that the withdrawal of stabilizing water molecules accounted for virus inactivation. A collapse of nucleic acid structure and irreversible secondary reactions provoked the decay of the whole virus particle. If the stabilizing water molecule is desiccated once under dehydrating conditions, it is less likely to be replaced by another one. A recently published review focused on the vulnerability of SARS-CoV-2 regarding NaCl and presented this in detail [8]. This type of antiviral activity, utilized against a wide range of viral infections, can be augmented by the availability of NaCl [9,10]. Virus particles are sensitive to dehydrating conditions (as exerted by salt) because they need structurally unmodified surface proteins and nucleic acids for the binding and infection of target cells. Hypertonic conditions on face masks facilitate the denaturation of microorganism proteins, and consequently prohibit replication of these pathogens [5,6,11]. Moreover, chloride ions are utilized by cells to exert antiviral activity through the production of hypochlorous acid. As previously discussed [5,6], NaCl would be a first-line candidate for mask impregnation and an important protection strategy for the global population.

Against this background, this study aimed to prove that hypertonic saline effectively reduces the infectious viral load on treated masks. In this respect, we hypothesized that hypertonic saline significantly reduces viral activity on impregnated FFPs while also ensuring long-term germless conditions in the presence of air. Therefore, we investigated both the infectivity of TGEV after contact with hypertonic saline-coated FFPs in a porcine testicular cell line (determined as the most appropriate cell line for this virus) [12,13], and the effect of salt solutions on airborne microorganisms. TGEV is a highly contagious enteric viral disease in swine, characterized by vomiting, severe diarrhea, and high mortality (often 100%) in piglets less than two weeks of age. TGEV was used as a surrogate of the SARS-CoV in previous studies, as it showed longer survival in water compared to the mouse hepatitis virus, a *Betacoronavirus*, which belongs to the same genus as SARS-CoV and SARS-CoV-2 [14]. There is a high probability that TGEV is more resistant to environmental conditions than SARS-CoV-2, and it can be used safely in such experiments. Thus, the efficiency of hypertonic saline in the inactivation of TGEV should be indicative of a pronounced anti-SARS-CoV-2 effect as well.

## 2. Methods

If not indicated otherwise, all chemicals were obtained from Merck (Darmstadt, Germany) and Sigma-Aldrich (St. Louis, MO, USA). MIHESA^®^ (sodium chloride solution 10%) was received from Omnignostica Ltd. (Höflein/Danube, Austria).

### 2.1. Virus and Cell-Culture

Swine testicular (ST) cells were cultured at 37 °C in an atmosphere of 5% CO_2_. Dulbecco´s Modified Eagle´s Medium (DMEM) culture medium (Lonza, Walkersville, MD, USA) was supplemented with 10% fetal bovine serum (FBS, Biowest, Nuaille, France) and antibiotic-antimycotic solution (Sigma-Aldrich, St. Louis, MO, USA). Porcine Alphacoronavirus 1 (TGEV) was stored at −80 °C before propagation on ST-cells. A standardized infectious titer of 10^6.5^ TCID_50_ (50% tissue culture infectious dose) per milliliter was established for the experiments by dilution, based on the stock virus´s titration results. 

In the provision of research involving cell lines, the ST cell line was kindly provided by Dr. Attila Cságola (current affiliation: Ceva Phylaxia Ltd., Budapest, Hungary).

### 2.2. Experimental Design

Standard high-quality surgical masks were treated in an aseptic environment as follows:

1. Infectivity of TGEV in case of untreated surgical masks (Experiment-1).

The outer layer of surgical masks was sprayed with TGEV. This process began with five pump-strokes from a 100 mL spray flask, at a distance of 20 cm, with TGEV virus suspension of 10^6.5^ TCID_50/mL_, which resembled true-to-life conditions. After a drying time of six hours, a square of 3 × 3 cm was cut from the central part of each mask and incubated for another 16 h in 5 mL DMEM at room temperature. The eluate was then titrated in quadruplicates by performing a 10-fold serial dilution in DMEM on a 96-well microtiter plate (TPP, Trasadingen, Switzerland), followed by the addition of ST-cell suspension (4 × 10^4^ cells/well). Titration plates were then incubated for six days at 37 °C and observed daily for virus-related cytopathogenic effects (CPE), recorded as cell-rounding, cell-fusion, and cell-death;

2. Infectivity of TGEV in primarily untreated surgical masks and subsequent treatment with sodium-chloride (Experiment-2).

Repeating the previous process, the outer layer of surgical masks was sprayed with five pump-strokes from a 100 mL spray flask, at a distance of 20 cm, with TGEV 10^6.5^ TCID_50/mL_. After a drying time of three hours, the infected masks were treated with NaCl (10%) and dried for another three hours. Further treatment was performed as described above;

3. Infectivity of TGEV concerning the NaCl pre-coated masks (Experiment-3).

Surgical mask surfaces (outer and inner layers) were pre-impregnated with 10% sodium chloride for nine hours preceding contact with TGEV. After desiccation, surgical masks were treated according to the procedure as described in experiment 1;

4. Treatment of surgical masks with DMEM as control (Experiment-4).

The outer layer of untreated surgical masks was sprayed with five pump-strokes from a 100 mL spray flask, at a distance of 20 cm, with DMEM. After a drying time of six hours, a square of 3 × 3 cm was cut from the central part of each mask and incubated for another 16 h in 5 mL DMEM. Further treatment was performed as described above. 

### 2.3. Effect of Salt Solutions on Airborne Microorganisms

Sodium-chloride in mg/mL (0 = control, 10, 25, 50 and 100) was dissolved in phosphate-buffered saline (PBS) containing 5% saccharose. A total of 10 mL of each concentration was filled in vials and positioned without cover for 8 h to facilitate infection in the sodium-chloride solutions with airborne microorganisms. Next, the vials were covered slightly to enable a constant oxygen supply and left at room temperature for 100 days. At the end of the incubation time, contamination of these salt solutions with micro-organisms was detected turbidimetrically by measuring optical density at 650 nm.

### 2.4. Statistics

Statistical analysis was performed using the Sigma-Plot package 14.0 (SPSS, Erkrath, Germany). Tests for normal distribution were performed with the Shapiro–Wilk test. Differences between treatments were analyzed with the Kruskal–Wallis one-way analysis of variance on ranks; *p* values of <0.05 were considered to be significant. 

## 3. Results

### 3.1. Infectivity of TGEV in STE62 Cells after Contact with the Surface of Surgical Masks

In the first experimental approach, when TGEV was applied on surgical masks without hypertonic saline impregnation, viral load was scarcely reduced, i.e., a reduction in the initial titer from 10^6.5^ TCID_50_/_mL_ to a mean of 10^5.86^ TCID_50/mL_. Cytopathogenic effects of the virus on STE62 cells are exemplified in Figure 1.

In the second attempt, when the infected surgical masks were dried for three hours and submitted to a post-treatment with hypertonic saline for another three hours until extraction of the TGEV in DMEM, a significant decrease in the viral load by a factor of >10^4^ was observed, i.e., a reduction in the initial titer from 10^6.5^ TCID_50/mL_ to a mean of 10^2.36^ TCID_50/mL_ as indicated in Table 1.

The most potent antiviral effect was achieved on the hypertonic saline pre-coated surgical masks. The viral load decreased from an initial titer of 10^6.5^ TCID_50/mL_ to a mean of 10^1.74^ TCID_50/mL_ (see Figure 2).

### 3.2. Growth Inhibition of Hypertonic Salt Solutions on Airborne Microorganisms

The spontaneous inoculation of airborne microorganisms in the culture medium was significantly inhibited in the presence of salt, with a critical range of five percent NaCl, as indicated in Figure 3.

Spontaneous inoculation of airborne microorganisms in the medium was observed for a period of 100 days. In the presence of ≥5% sodium chloride, the growth of microorganisms was significantly inhibited. Growth inhibition upon increasing hypertonic saline was determined through turbidity measurement at 650 nm. Data were plotted as mean value ± standard deviation.

## 4. Discussion

In the present study, we investigated the protective effect of hypertonic salt solutions coated on FFPs. To reduce the infection risk for the technical staff involved in this study, TGEV, belonging to species Alphacoronavirus 1, was used as a model to test filtration efficiency against viral aerosols as a surrogate in place of the pathogenic SARS-CoV-2. In line with previous studies (involving bacteria and *Saccharomyces cerevisiae* [5,6]), we observed a concentration-dependent growth inhibition of microorganisms by hypertonic salt solutions. This study’s most important findings are that salt per se is a life-hostile environment for airborne microorganisms. Infectious viral load on FFPs was significantly reduced by hypertonic-salt impregnation before and after contamination with TGEV, disproving the null hypothesis.

As virus-containing aerosols spread worldwide, FFP respirators are essential tools to protect against airborne viral particles. Diverse FFP devices are commercially available and are made of cellulose or cotton materials and synthetic and nanotechnological fibers. FFP devices comprise several layers, including antimicrobial layers, such as silver-copper zeolite, divalent metal cations, and citric acid, which are coated on or embedded between the layers. The treatment of FFP respirators with suspensions of dialdehyde starch increased the virucidal efficacy significantly and did not modify the virus removal efficiency or pressure drop of the filter [15,16]. Surgical masks were envisaged to save susceptible persons from the wearer’s exhaled breath compared to surgical respirators, preventing infections of the wearer’s respiratory system [17]. However, in a previously published report, high-quality standard surgical masks were reported to be as effective as FFP2 masks, although neither mask type completely prevents transmission. The incorrect donning and doffing of masks and lack of hand hygiene may cause contagion by the wearer’s own hands through contaminated touchable surfaces, especially the mask’s outer layer. The adhesion of microorganisms is mainly facilitated as a result of van der Waals interaction on the flat surface of masks [18,19,20]. This may be of particular interest concerning virus re-aerosolization during extended use of masks due to an accumulation of viral aerosols at the surface and particle migration within the filtration of the respirators, where they survive for an extended period [15]. During the outbreak of severe acute respiratory syndrome (SARS), medical face masks were indispensable to protect against infection. The use of masks was associated with a 12.6 times risk of mitigation compared to subjects without mask-wearing [21].

During a speech, an average emission of 1000 and up to 10,000 droplet particles with high emission rates was determined. Loudness is directly proportional to droplet particle emission, i.e., ranging from six particles per second while whispering to 53 particles per second when speaking loudly. There is also a significant variability between individuals, with super-spreaders being particularly contagious [22]. Moreover, five coughs of a COVID-19 patient, within a distance of 20 cm, in front of a petri dish, reduced the viral copies from 2.6 log^10^ per mL without a mask to 2.4 log^10^ with a surgical mask, as reported previously [23]. SARS-CoV-2 spreaders contaminated the face mask’s outer layer with only five coughs with viable viruses for up to seven days and four days on the inner layer. First and foremost, commercially available FFPs capture viral particles but do not inactivate microorganisms. To reduce the risk of self-contagion, FFPs must be changed frequently, contributing to environmental pollution worldwide, since most masks are made of non-biodegradable and non-renewable polymers. The incipient stages of solving these problems are using multi-use biodegradable masks, e.g., hybrid masks, incorporating non-woven cellulose fibre, natural extracts from green tea, sialic acid, or impregnation with copper oxide [24]. Mask decontamination abilities are diverse and range from ultraviolet radiation, hydrogen peroxide vapour, heat, and moisture to disinfectant solution treatments, each presenting a variety of advantages and drawbacks [22,25,26]. As presented in the current study, hypertonic salt solutions have a life-hostile impact on airborne microorganisms. Salt solutions greater or equal to 5% NaCl could inhibit colonization and growth of airborne microorganisms in the culture medium. This is consistent with a previous report where yeast cell-growth was inhibited by the same salt concentration, i.e., a unicellular organism that comprises a defending cell wall, in contrast with viruses [6]. Moreover, there was no significant reduction in virus titers recovered from untreated control surgical masks, which indicates that bare FFP did not influence the infectivity of the virus. In this context, it should be mentioned that the treatment of masks with 10% NaCl resulted in a >99.99% reduction in porcine Coronavirus infectivity. The sequence of mask impregnation and inoculation with viruses was of less importance. However, the virus titer’s decrease in invectivity was more pronounced when the FFP was pre-coated with 10% NaCl before virus inoculation. These results are concurrent with a previous publication, where bacteria recovered from bare membranes, which were further used for infection in mice, induced a significantly higher bodyweight-loss compared to bacteria from membranes coated with 10% salt [5]. Interestingly, the same group reported that hypertonic saline coating induced an increased anti-microbial effect of filters at harsh conditions. A humid environment, similarly to a prolonged period of use, and unlikeuncoated membranes resulted in trapped humidity and provided a protected and increased growth in microorganisms. The top-layer (i.e., the salt-impregnation) is mainly responsible for the increased filtration capacity of salt-coated filters compared to bare membranes, indicating an overall enhancement in performance, ensuring a maximum of filtration efficiency with a negligible pressure drop. Additionally, this enables the functionalization of large-pore membranes, independent of layer-thickness, without increasing airflow resistance [5]. As a result of this study, it should be noted that the wearer of face masks with salt-impregnation may use the same mask for an extended period of time and will be efficiently protected from self-contagion by incorrect donning and doffing. A further benefit is the reuse of masks through a post-treatment with hypertonic-saline, i.e., at least once or twice; however, this should not be exceeded for hygienic reasons. In addition, multiple treatments may inhibit air permeability and cause skin irritations. Moreover, the wearing of a NaCl-coated mask may mimic the effects of dry salt inhalation (halotherapy), frequently used to prevent and treat respiratory tract infections. Due to the increased osmotic pressure of inhaled NaCl, the following effects were observed, which are likely to be of clinical benefit in times of SARS-CoV-2: a decrease in inflammation and bronchial hyperreactivity, increased mucociliary elimination, dissolution of mucus, inhibition of bacterial growth, an increase in phagocyte activity and effects on general well-being and quality of life, such as relaxation effects on the central nervous system [27,28,29]. Finally, it should be noted that salt impregnation of face masks is by no means specific, but likely affects all kinds of microorganisms. Therefore, the strategy of putting NaCl on mask surfaces should perhaps not be restricted to the COVID-19 pandemic, but used as a precaution in general, and may also help to reduce associated oxidative stress.

## 5. Conclusions

In sight of the COVID-19 pandemic, and knowing that conventional face masks accumulate pathogens, FFPs coated with hypertonic saline were shown to significantly reduce the infectivity of a porcine coronavirus (TGEV). This is attributable to the salt-layer, provided that salt per se is a life-hostile environment for microorganisms. In this respect, hypertonic salt solutions coated on FFPs reduce the risk of infection. Hypertonic saline impregnation has a wide range of advantages: it is simple, affordable, user- and eco-friendly, resource sparing, and can be easily performed by everyone for each kind of mask. Moreover, the risk of self-contagion through the outer layer is significantly decreased, ensuring a maximization of protection, even with the consideration of low compliance with the appropriate utilization of face masks.

## Figures and Tables

**Figure 1 ijerph-18-07406-f001:**
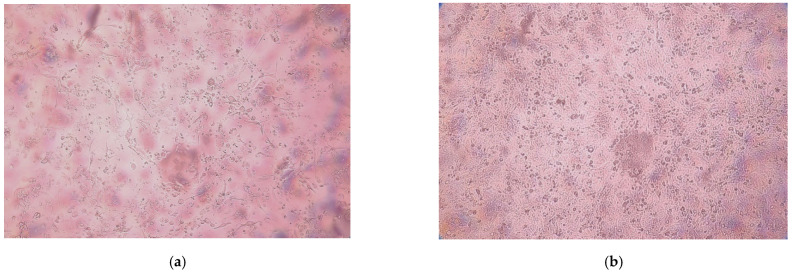
Cytopathogenic effects of TGEV on STE62 cell cultures (**a**) in comparison to the control culture (**b**) three days after inoculation. Inverted microscope Optika XDS-2 (Optika, Ponteranica, Italy), 100× magnification.

**Figure 2 ijerph-18-07406-f002:**
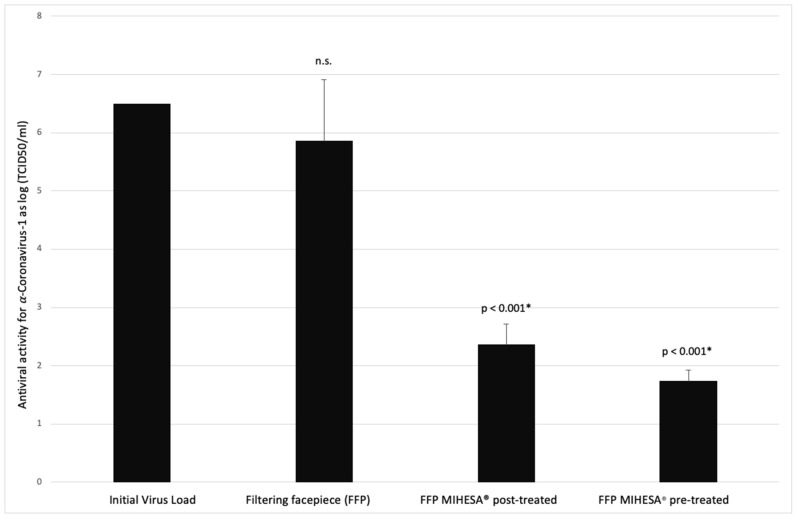
TGEV inhibiting activity (initial titer = 10^6.5^ TCID_50_/mL) on bare FFP compared to FFP’s treated with hypertonic saline (MIHESA). n.s. = not significant; *: indicate the significance level.

**Figure 3 ijerph-18-07406-f003:**
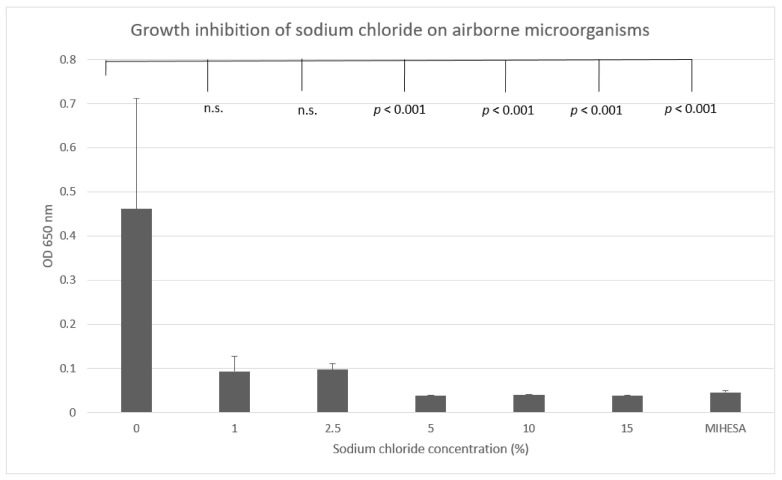
Growth inhibition of sodium chloride on airborne microorganisms. OD: OD = optical density; n.s. = not significant.

**Table 1 ijerph-18-07406-t001:** Antiviral activity of hypertonic saline, coated on surgical masks.

Titer		Exp-1	Exp-2 *	Exp-3 *
Initial	(TCID_50_/mL)	10^6.5^	10^6.5^	10^6.5^
Mean	(TCID_50_/mL)	10^5.86^	10^2.36^	10^1.74^
SD		10^1.05^	10^0.36^	10^0.18^
Min	(TCID_50_/mL)	10^5.12^	10^2.11^	10^1.62^
Max	(TCID_50_/mL)	10^6.61^	10^2.62^	10^1.87^
Delta	(TCID_50_/mL)	10^−0.64^	10^−4.14^	10^−4.76^

Exp-1 represents the viral load of Transmissible Gastroenteritis Virus (TGEV) (initial titer = 10^6.5^ TCID_50/_mL – 50% tissue culture infectious dose) sprayed on bare surgical masks with subsequent elution in Dulbecco’s Modified Eagle Medium (DMEM) and inoculation with ST cells. The titer after elution is indicated as mean, standard deviation (SD), minimum, maximum, and delta compared to the baseline titer. In Exp-2, TGEV was sprayed on bare surgical masks and post-treated with hypertonic saline three hours after infection. In Exp-3, surgical masks were coated with hypertonic saline 9 h in advance of infection with TGEV. *: indicate a significant reduction in viral activity (*p* ≤ 0.001) compared to bare surgical masks without hypertonic saline.

## Data Availability

The data used to support the findings of this study are available from the corresponding author upon request.

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
