# Peer review of "Coating with Hypertonic Saline Improves Virus Protection of Filtering Facepiece Manyfold—Benefit of Salt Impregnation in Times of Pandemic"

_ijerph, 2021, doi:10.3390/ijerph18147406_

Round 1

Reviewer 1 Report

In this study, the authors discover the protective effect of a hypertonic salt solution on the infectivity of TGEV on surgical mask, and also access the long-term infectivity of airborne microorganism of NaCl in PBS with 5% saccharose. The results revealed a significant reduce of TCID50 of TGEV by treated with in FFP pre-coated with hypertonic saline, and an obvious inhibition of airborne microorganism when salt concentrations over 5%. The hypertonic saline is known to a well approach to prevent viral infection or replication, but it is a novel tool for useful for anti-viral protection in emerging infectious diseases. There are some flaws in the manuscripts to be solved to meet the requirements of readership.

(1) In Table 1, it would be a better way to present the data visually in a graph, rather than a table. The numbers with exponents could be display in a form of log value in y axis.

(2) In Discussion section, the text Line 203-258 is rigmarole. It needs to be rearranged and make it logical to increase the readership.

Author Response

Point to point reply - Reviewer 1:

There are some flaws in the manuscripts to be solved to meet the requirements of readership.

R: In Table 1, it would be a better way to present the data visually in a graph, rather than a table. The numbers with exponents could be display in a form of log value in y axis.

A: We followed the proposed amendment of the reviewer and included figure 2 as numbers with exponents displayed as log values in y-axis.

R: In Discussion section, the text Line 203-258 is rigmarole. It needs to be rearranged and make it logical to increase the readership.

A: We thank the reviewer for this vigilant annotation. We have rearranged this section between lines 242 and 259 to make the discussion easier to read coherently.

Reviewer 2 Report

Title: Coating with hypertonic saline improves virus protection of filtering facepiece manyfold - benefit of salt impregnation in times of pandemic

ijerph-1245445

The aim of the present investigation was to evaluate the hypertonic saline coating for virus protection of filtering facepiece devices.

GENERAL COMMENTS

The article is original and the methods and study outcome is clearly written and discussed in the main text. The present manuscript should solve methodologic issues to be considered for publication.

Abstract

The aim of the study statement is missed and should be introduced.

Introduction

The aim of the study statement is missed and should be introduced.

The swine testicular (ST) cells and TGEV model was not justificated adequately in the introduction section. It is not clear the affinity with the respiratory disease and the SARS-CoV-2 viral vector, and the translational value with the human model.

Materials and methods

The analytic methods are completely missed in the section. What did you evaluated? PCR? And genes expression? Cells differentiation? Cells proliferation? No information about the environmental condition has been introduced (temperature, humidity…) about the surgical mask experiment.

For the present investigation probably a immunofluorescence approach seems to be more appropriate to support the authors hypothesis.

Discussion:

No mentions about the surgical mask composition and molecular. The null hypothesis discussion is missed in the present paragraph.

Author Response

Point to point reply – Reviewer 2:

Reviewer 2:

R: Abstract

The aim of the study statement is missed and should be introduced.

A: We want to thank the reviewer for this attentive annotation. We followed the suggestion of the reviewer and included the following sentence in the Abstract (line 19): “This study aimed to prove that hypertonic saline effectively reduces the infectious viral load on treated masks.”

R: Introduction

The aim of the study statement is missed and should be introduced.

A: We followed the reviewer´s suggestion and stated the aim of the study in the “Introduction” section at line 74. “Against this background, this study aimed to prove that hypertonic saline effectively reduces the infectious viral load on treated masks. In this respect, we …“

R: The swine testicular (ST) cells and TGEV model was not justificated adequately in the introduction section. It is not clear the affinity with the respiratory disease and the SARS-CoV-2 viral vector, and the translational value with the human model.

A: Swine testicular cells are the most widely used and probably the most appropriate cell lines for TGEV propagation (https://doi.org/10.1371/journal.pone.0110647; https://doi.org/10.1016/j.rvsc.2015.09.009). We included this detail in the Introduction at line 79 comprising the respective citations.

The laboratory use of zoonotic coronaviruses is restricted to the BSL-3 safety level, which makes the use of surrogate viruses necessary for the investigations. In this respect we used TGEV as a model virus for SARS-CoV-2. Although TGEV is a coronavirus that causes an enteric disease, it can serve as a model organism in the current study, which does not involve the airborne infection event itself, but only concentrates on virus survival (infectivity of the virus particles) on the surface of the FFPs. In this regard, TGEV is attributed with greater environmental resistance than coronaviruses, therefore the inactivation of TGEV is a higher challenge to the tested system than that of SARS-CoV-2. Thus, the proven efficiency of hypertonic saline in the inactivation of TGEV is most likely indicative of a pronounced anti-SARS-CoV-2 effect as well. In this respect, we amended the last sentence in the Introduction (line 86).

The translational value of the test system to a human situation - in our interpretation, the fact that coating with hypertonic saline significantly reduces the infectious viral load on the FFPs is a clear indicator of a decreased infection risk even in a real-world scenario. Infectious viruses were transferred to the surface of FFPs using a spray pump that is similar to the real spread of aerogene viruses by airborne droplets. Of course, field experiments with people using the coated masks in an epidemic environment could provide further proof of concept, but the design and implementation of such a study would be rather complicated.

R: Materials and methods

The analytic methods are completely missed in the section. What did you evaluated? PCR? And genes expression? Cells differentiation? Cells proliferation? No information about the environmental condition has been introduced (temperature, humidity...) about the surgical mask experiment.

A: The aim of the experiment was to prove that coating with hypertonic saline effectively reduces the infectious viral load on treated masks. Therefore, we had to detect only the amount of the infectious, replication-competent virus particles (not including the inactivated virions that pose no infection threat), to which only an evaluation method based on virus propagation is suitable. Virus titration in cell culture is a classic virology method that relies on the end-dilution of virus-containing sample, inoculation of parallel cell cultures and statistical calculation (Spearman-Kärber or Reed-Munch method) for the quantification of infectious particles. That method was used for the evaluation of the results in the surgical mask experiment. Any other analytic method (like measuring virus-specific RNA with qRT-PCR or virus-specific antigens with immunological-based methods) could only be used as a supplement; alone they would have lead to misleading results (as they cannot differentiate between infective and inactivated viruses).

R: For the present investigation probably a immunofluorescence approach seems to be more appropriate to support the authors hypothesis.

A: We want to thank the reviewer for addressing this decisive point. An immunofluorescence approach is always a good option to help in the better visualization and more specific detection of infected cells in cell culture, but in this study, we considered it unnecessary as the cytopathogenic effects caused by TGEV were clearly visible and specific enough for the evaluation of viral titers.

R: Discussion:

No mentions about the surgical mask composition and molecular. The null hypothesis discussion is missed in the present paragraph.

A: We mentioned the type of mask in the “Material and Methods” section at line 120: “Standard high-quality surgical masks were treated …”.

The mask is commercially available safety protection with a three-layer filtration (two layers of polypropylene non-stick non-woven fabric and one layer of melt-blown non-woven fabric), BFE>95%, with medical sterility, odor absorption, light and breathable, comfortable to wear, authoritative certificated, safe and reliable, with a mask dimension of 17x9.5cm and a mask weight of 2.8g/pc. In this context, it should be emphasized that neither the type of mask nor the number of layers is decisive for the protective effect of the hypertonic salt solution. As already published by Rubino et al. 2020 – (“This led us to hypothesize that the top layer (first layer to interact with the aerosols) is mainly responsible for the increased filtration efficiency observed in salt-coated filters.” Ref.: Sci Rep 10, 13875, 2020; https://doi.org/10.1038/s41598-020-70623-9) as cited in this manuscript at line 277 ff.), “The top-layer (i.e., the salt-impregnation) is responsible for the increased… .” 

The null hypothesis was disproved by the results. This was highlighted at line 213: “… with TGEV, that disproved the null hypothesis.”

Reviewer 3 Report

This is an interesting manuscript regarding the influence of FFP coating with hypertonic saline on the infectiousness of alphacoronavirus 1. The manuscript is written in a sound manner. I have no comments regarding English Language. I highly appreciate that the study also proposed practical implications which are quite optimistic.

I have several comments:

In the Introduction or Discussion section you have briefly mentioned certain procedures that can impede the true protective value of face masks. I would consider adding some citations of studies which demonstrated that usage of face masks is improper according to the official WHO criteria. Not only is this the issue with the general population, but with medical professionals as well. For example, check papers from IJERPH like 10.3390/ijerph17186484 or 10.3390/ijerph18020841 

Conclusion section:  "In this respect, hypertonic salt solutions coated on FFPs reduce the risk of infection in the present COVID-19 pandemic". In the manuscript you have focused on TGEV and provided a rationale that SARS-CoV-2 is even easier to destroy. Still, the underlined fragment presented in the Conclusion should be less definite. Your study wasn't performed on SARS-CoV-2, so you are extrapolating the data. 

"As a result of this study, it should be noted that the wearer of face masks with salt-impregnation may use the same mask for an extended time and will be efficiently protected from self-contagion by incorrect donning and doffing." - for how long? One of the basics of face mask use is frequent replacement and disposing of the used ones. Paradoxically, could this sense of safety turn out to be false after some period of time? Basically, the accumulation of moisture due to continuous use is a serious factor contributing to the growth of bacteria. I imagine there could be a turning point in the continuous use of salt-coated masks when these protective properties fail and from then on it would become a microbiological "bomb". Not to mention that prolonged use may be associated with several cutaneous inconveniences. 

Author Response

Point to point reply – Reviewer 3:

Reviewer 3:

I have several comments:

R: In the Introduction or Discussion section you have briefly mentioned certain procedures that can impede the true protective value of face masks. I would consider adding some citations of studies which demonstrated that usage of face masks is improper according to the official WHO criteria. Not only is this the issue with the general population, but with medical professionals as well. For example, check papers from IJERPH like 10.3390/ijerph17186484 or 10.3390/ijerph18020841

A: We would like to thank the reviewer for this valuable hint. We incorporated these two references into the Introduction as references 3 and 4.

R: Conclusion section: "In this respect, hypertonic salt solutions coated on FFPs reduce the risk of infection in the present COVID-19 pandemic". In the manuscript you have focused on TGEV and provided a rationale that SARS- CoV-2 is even easier to destroy. Still, the underlined fragment presented in the Conclusion should be less definite. Your study wasn't performed on SARS-CoV-2, so you are extrapolating the data.

A: We want to thank the reviewer for this attentive comment. In the Conclusions, we removed the phrase (line 373), “…in the present COVID-19 pandemic. ” Moreover, the last sentence (line 376) was amended through: “… downscaled, ensuring maximization of protection even under the aspect that the compliance for appropriate utilization of face masks is rather low.”

R: "As a result of this study, it should be noted that the wearer of face masks with salt-impregnation may use the same mask for an extended time and will be efficiently protected from self-contagion by incorrect donning and doffing." - for how long? One of the basics of face mask use is frequent replacement and disposing of the used ones.
Paradoxically, could this sense of safety turn out to be false after some period of time? Basically, the accumulation of moisture due to continuous use is a serious factor contributing to the growth of bacteria. I imagine there could be a turning point in the continuous use of salt-coated masks when these protective properties fail and from then on it would become a microbiological "bomb". Not to mention that prolonged use may be associated with several cutaneous inconveniences.

A: We thank the reviewer for this extremely mindful comment. We completely agree that a long wear time makes untreated FFP ineffective and will ultimately lead to a microbial bomb. On the contrary, the paper of Rubino et al. (Ref.5) presented the effect of a hypertonic salt impregnated FFP under harsh conditions (i.e., 37°C 90 % relative humidity). “… the salt coated filters showed a significant increase in inactivation properties after exposure to the humid environment, and no bacteria were detected from the filters after storage.” Repeated application with hypertonic salt solutions renders the masks impermeable apart from cutaneous inconveniences. Thus, the impregnation should not exceed one to two applications.  We have done justice to this detail by inserting the following sentence (line 353): “A further benefit is the reuse of masks through a post-treatment with hypertonic-saline, i.e., at least once or twice, but should not exceed this for hygienic reasons. In addition, multiple treatments may inhibit air permeability and cause skin irritations.”

Round 2

Reviewer 2 Report

The authors solved correctly the previous critical query.

The manuscript is now suitable for pubblication.